# The impact of earthquake cycle variability on neotectonic and paleoseismic slip rate estimates

**Richard Styron**[1, 2, 3]

[1]Earth Analysis, 21855 Bear Creek Road, Los Gatos, CA 95033 USA
[2]Global Earthquake Model Foundation, Via Ferrata 1, Pavia 27100 Italy
[3]Department of Geology, University of Kansas, Ritchie Hall, Earth Energy & Environment Center, 1414 Naismith Drive, Room 254, Lawrence, KS 66054 USA

**Correspondence:** Richard Styron (richard.h.styron@gmail.com)

**Abstract.** Because of the natural (aleatoric) variability in earthquake recurrence intervals and coseismic displacements on a fault, cumulative slip on a fault does not increase linearly or perfectly step-wise with time; instead, some amount of variability in shorter-term slip rates results. Though this variability could greatly affect the accuracy of neotectonic (i.e., late Quaternary) and paleoseismic slip rate estimates, these effects have not been quantified. In this study, idealized faults with four different, representative earthquake recurrence distributions are created with equal mean recurrence intervals (1,000 years) and coseismic slip distributions, and the variability in slip rate estimates over 500 to 100,000 year measurement windows is calculated for all faults through Monte Carlo simulations. Slip rates are calculated as net offset divided by elapsed time, as in a typical neotectonic study. The recurrence distributions used are quasi-periodic, unclustered and clustered lognormal distributions, and an unclustered exponential distribution. The results demonstrate that the most important parameter is the coefficient of variation ($CV$ = standard deviation / mean) of the recurrence distributions rather than the shape of the distribution itself. Slip rate variability over short time scales (<5,000 years or 5 mean earthquake cycles) is quite high, varying by a factor of 3 or more from the mean, but decreases with time and is close to stable after ~40,000 years (40 mean earthquake cycles). This variability is higher for recurrence distributions with a higher $CV$. The natural variability in the slip rate estimates compared to the true value is then used to estimate the epistemic uncertainty in a single slip rate measurement (as one would make in a geological study) in the absence of any measurement uncertainty. This epistemic uncertainty is very high (a factor of 2 or more) for measurement windows of a few mean earthquake cycles (as in a paleoseismic slip rate estimate), but decreases rapidly to a factor of 1—2 with >5 mean earthquake cycles (as in a neotectonic slip rate study). These uncertainties are independent of, and should be propagated with, uncertainties in fault displacement and geochronologic measurements used to estimate slip rates. They may then aid in the comparison of slip rates from different methods or the evaluation of potential slip rate changes over time.

## 1 Introduction

Fault slip rates are generally estimated by dividing measurements of the offset of geologic marker features by the time over which that offset accumulated (it is not currently possible to measure a slip rate directly, though the term 'slip rate measurement' may be used to compare to a simulated or modeled value). The uncertainty in the resulting slip estimate is typically treated as *epistemic*, and quantified through the propagation of the measurement uncertainties on the offset and time quantities (e.g., Bird, 2007; Zechar and Frankel, 2009). However, for slip rate estimates on active faults made from offset measurements near the fault trace (i.e., within a horizontal distance that is a small fraction of the fault's locking depth, as the width of the zone affected by earthquake-cycle strains is a function of locking depth (e.g., Savage and Burford, 1973; Hetland and Hager, 2006)), the episodic nature of surface displacement due to the fault's position in the earthquake cycle will necessarily affect the results: If the measurements are taken immediately before an earthquake, the measured offset and resulting slip rate estimate will be lower than average, while if the measurements are taken im-

mediately after an earthquake, the offset and rate will be higher.

The magnitude of the perturbation to the slip rate estimate is, of course, a function of the number of cumulative earthquakes that have contributed to the measured offset (plus any aseismic strain such as afterslip or creep). For older Quaternary markers that have experienced tens to hundreds of major earthquakes, the effects will be minor, and for bedrock geologic markers with kilometers of displacement, the earthquake cycle is likely not worth accounting for. However, due to progressive erosion of geologic markers and the challenge of dating many late Pliocene to early Quaternary units (which are too old for radiocarbon and many cosmogenic nuclide systems), geologists often have no choice but to choose late Pleistocene to Holocene markers to date. These units may also be more desirable targets if the scientists are primarily concerned with estimating the contemporary slip rate on a fault with a slip rate that may vary over Quaternary timescales (e.g., Rittase et al., 2014; Zinke et al., 2018). For slow-moving faults, the slip either long waiting to be released, or recently released, may represent a sizeable fraction of the measured fault offset.

Careful paleoseismologists and neotectonicists will take this into account in their slip rate calculations if sufficient data are available, especially in the years after a major earthquake (e.g., Rizza et al., 2015), and many others will discuss the potential effects if the data are not (e.g., Lifton et al., 2015). These workers may only consider the time since the last earthquake, often making the assumption (stated or not) that the earthquakes are identical in slip and perfectly periodic.

However, the recurrence intervals between successive earthquakes on any given fault segment have some natural (i.e. *aleatoric*) variability; similarly, displacement at a measured point is not identical in each earthquake (e.g., DuRoss, 2008). Therefore, the measured slip rate may deviate from the time-averaged rate based on the amount of natural variability in the earthquake cycle, particularly given successive events from the tails of the recurrence interval or displacement distributions.

The physical mechanisms responsible for the aleatoric variability in earthquake recurrence intervals and displacements are still unclear, and the subject of active investigation. Most earthquakes serve to release differential stresses caused by relative motions of tectonic plates or smaller crustal blocks; relative plate velocities measured over tens of thousands to millions of years from geologic reconstructions are similar enough to those measured over a few years through GPS geodesy that sudden, transient accelerations and decelerations are unlikely (e.g., DeMets and Dixon, 1999). As a consequence, plate boundary faults may have near-constant rates of loading from tectonic stress. Many plate boundary faults are are among the most regularly-rupturing faults known, particularly sections that are isolated from nearby faults (e.g., Berryman et al., 2012) and therefore not affected by stress perturbations resulting from earthquakes on other faults.

These stress perturbations may be 'static' coseismic instantaneous stresses in the elastic upper crust resulting from earthquake displacement (King et al., 1994), or analogous post-seismic stress changes in the viscoelastic lower crust or upper mantle from the time-dependent relaxation of static stress perturbations (e.g., Chéry et al., 2001), or 'dynamic' transient stress changes that accompany the passage of seismic waves from nearby or distant earthquakes (Gomberg and Johnson, 2005). Additionally, changes in pore fluid pressure in a fault zone may increase or decrease the required shear stress to initiate an earthquake (Steacy et al., 2005). In contrast to isolated plate boundary faults, intraplate faults or those on distributed plate boundaries may have lower stress accumulation rates and the stress perturbations from activity on other faults may be enough to significantly affect the timing of earthquakes on a given fault (Gomberg, 2005).

Though the physical mechanisms responsible and the statistical character of this natural variability remain under debate, its effects on the estimated slip rates may still be estimated given some common parameterizations.

In this study, the effects of the natural variability in earthquake recurrence intervals and per-event displacements on neotectonic slip rate estimates are investigated through Monte Carlo simulations. The study is geared towards providing useful heuristic bounds on the aleatoric variability and epistemic uncertainty of late Quaternary slip rate estimates for fault geologists, probabilistic seismic hazard modelers, and others for whom such uncertainties are important.

## 2  Modeling the earthquake cycle

To study the effects of the natural variability in the earthquake cycle on estimated slip rates, long displacement histories of a simulated fault with different parameterizations of the earthquake recurrence distribution will be created. Then, the mean slip rate over time windows of various sizes will be calculated from each of the simulated displacement histories, and the distribution in these results will be presented, representing the natural variability in this quantity.

To isolate the effects of the earthquake cycle from other phenomena that may affect slip rate estimates, this study does not attempt to model erosion, nor does it consider any measurement uncertainty in the age or offset of the faulted geologic markers; these quantities are assumed to be perfectly known. Additionally, though natural variability in per-earthquake displacement is included in the model, it is minor and the same for all recurrence distributions; though it is a random variable in the simulations, it is not an experimental variable. Furthermore, though the model has one length dimension (fault offset), it is still best thought of as a point (0-dimensional) model, as there is no spatial reference or along-strike or down-dip variability, and hence the magnitude of

each earthquake is undefined, and no magnitude-frequency distribution exists.

## 2.1 Earthquake recurrence interval distributions

There are a handful of statistical models for earthquake recurrence interval distributions that are under widespread consideration by the seismological community.

The most commonly used is the *exponential* distribution. This is associated with a Poisson process, and is the distribution that results from earthquakes being distributed uniformly randomly within some time interval. Consequently, the probability of an earthquake (or other event) occurring at any time does not change with time since the previous event (in other words, the hazard function is time-invariant); this leads to the characterization of the exponential recurrence distribution as 'random', 'memoryless' or 'time-independent'. The exponential distribution is also the simplest to describe statistically, as it requires only one parameter (the mean rate parameter), which is the inverse of the statistical scale parameter. (The scale parameter of a distribution determines the dispersion of the distribution, while the shape parameter determines the shape or form of the distribution; a parameter that shifts the distribution along the *x*-axis is called a location parameter.) The standard deviation of a large number of samples generated from an exponential distribution is equal to the mean.

The other distributions that are in common usage are time-dependent distributions, meaning that the probability of an event occurring at any time since the previous event changes with the elapsed time since that event. This class of distributions includes the lognormal, Weibull, and Brownian Passage Time (Matthews et al., 2002) distributions. Though these distributions differ in notable ways, particularly in the properties of the right tails at several times the mean (Davis et al., 1989; Matthews et al., 2002), they share a broadly general shape, and given suitable parameters, generated sample sets of small size may not be substantively different. In fact, the distributions are similar enough that it is difficult if not impossible to discriminate between them given realistic seismologic and paleoseismologic datasets (Matthews et al., 2002; Ogata, 1999). These distributions are described by both scale and shape parameters.

The behavior of these distributions and of empirical datasets may be characterized by the regularity of the spacing between events (i.e. the recurrence intervals): these may be periodic, unclustered (i.e., 'random'), or clustered. Assignment into these categories is typically done with a parameter known as the coefficient of variation, or $CV = \sigma/\mu$, where $\sigma$ is the standard deviation of the recurrence intervals, and $\mu$ is the mean recurrence interval.

Periodic earthquakes are those that occur more regularly than random, and have a $CV < 1$ (i.e. $\sigma < \mu$). These may be generated by any of the time-dependent distributions described above with suitable scale and shape parameters.

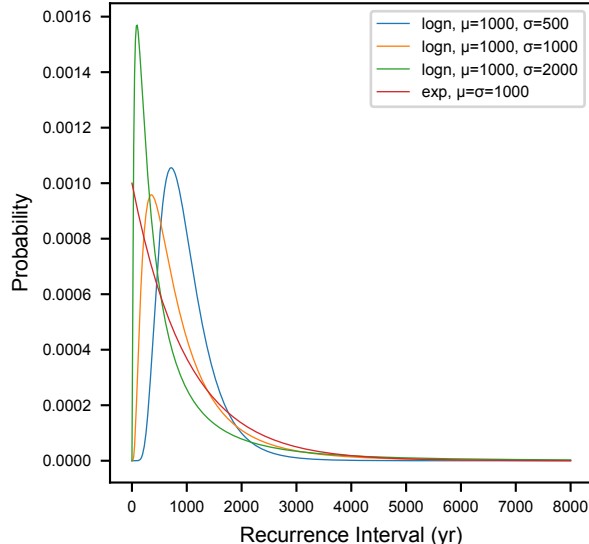

**Figure 1.** Earthquake recurrence distributions. 'logn' = lognormal. 'exp' = exponential. Colors for each distributions are the same in all figures.

(Note that in this paper, the use of the term 'periodic' does not mean *perfectly* repeating as it might in the physics or mathematics literature; the behavior referred to is most accurately termed 'quasi-periodic', but that term will not be used in the interests of conciseness.)

Unclustered earthquakes occur as regularly as random, and have a $CV = 1$. These may be generated by the exponential distribution (which can generate no other), or by any of the time-dependent distributions as well, given the appropriate parameters. Note that sample sets generated from these different distributions will not be identical, however: Sequences with an exponential recurrence distribution will have many more pairs of events that are much more closely spaced together than the mean, and more pairs of events that are much more widely spaced than the mean, compared to a sequence generated from the lognormal distribution. Nonetheless, these will cancel out in the aggregate statistics, so that the standard deviations will be equal. A comparison of these may be seen in Figure 1.

Clustered earthquake sequences have sets of very tightly spaced earthquakes that are widely separated (Figure 2, and have a $CV > 1$. These may be generated from a hyperexponential distribution, which is the sum of multiple exponentials with different means, or from the time-dependent distributions given above, given the right parameters.

No consensus exists among earthquake scientists as to the most appropriate recurrence interval distribution. As is generally the case with propriety, the safest and probably most correct assumption is that it is context-dependent. Many studies of plate boundary faults such as the San Andreas conclude

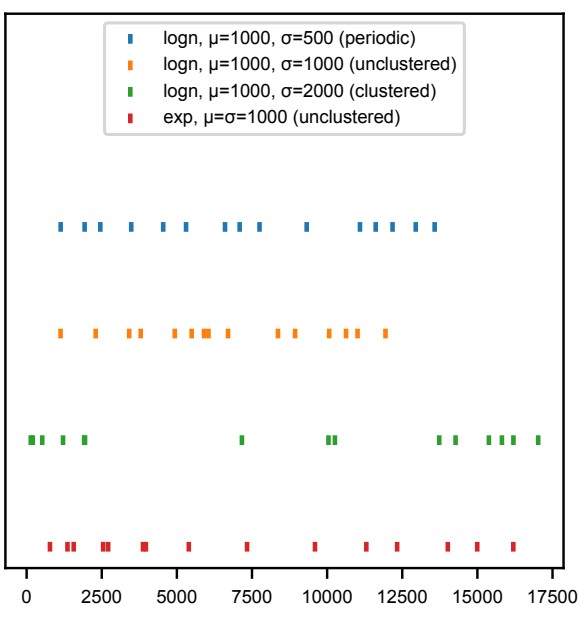

**Figure 2.** Spacing of 15 simulated successive earthquakes from each recurrence distribution. Note that the gap between the last displayed earthquake and the right side of the plot does not represent a long recurrence interval.

that major or 'characteristic' earthquakes are periodic (e.g., Berryman et al., 2012; Scharer et al., 2010). Conversely, many intraplate faults with low slip rates appear to show clustered earthquakes separated by long intervals of seismic quiescence (e.g., Clark et al., 2012). However, one can find examples of studies indicating the opposite conclusions, even from the same study areas (Tuttle, 2002; Grant and Sieh, 1994).

### 2.1.1 Modeled recurrence interval distributions

This study will compare four recurrence interval distributions (Figure 1):

1. A *periodic* distribution, represented by a lognormal distribution with a mean recurrence interval $\mu = 1000$ years, and a standard deviation $\sigma = 500$ years, and a $CV = 0.5$ [1]

---

[1]Please note that $\mu$ and $\sigma$ do not represent the scale and shape parameters of the lognormal distribution in this work, though these symbols are commonly used to represent these parameters elsewhere as the scale and shape parameters of a lognormal distribution $L$ are the standard deviation and mean of the normal distribution $\ln L$. Instead, the shape parameter $sh$ and scale parameter $sc$ are derived from $\mu$ and $\sigma$ through the relations $sh = \sqrt{1 + \sigma^2/\mu^2}$ and $sc = \ln[\mu/\sqrt{1 + \sigma^2/\mu^2}]$.

2. An *unclustered* time-dependent distribution, represented by a lognormal distribution with a mean recurrence interval $\mu = 1000$ years, a standard deviation $\sigma = 1000$ years, and a $CV = 1.0$.

3. An *clustered* time-dependent distribution, represented by a lognormal distribution with a mean recurrence interval $\mu = 1000$ years, a standard deviation $\sigma = 2000$ years, and a $CV = 2.0$.

4. An *unclustered* time-independent distribution, represented by an exponential distribution with a mean recurrence interval $\mu = 1000$ years, a standard deviation $\sigma = 1000$ years, and a $CV = 1.0$.

These distributions have been selected to represent a diversity of behaviors with a compact and tractable number of simulations, and particularly to explore how changes in $CV$ as well as the shape of the distribution impact slip rate estimates.

### 2.1.2 Earthquake slip distributions

All earthquake recurrence distributions share a single earthquake slip distribution (Figure 3). This distribution is a lognormal distribution with $\mu = 1$ m and $\sigma = 0.75$ m, which produces essentially 'characteristic' earthquakes that still nonetheless have some variability. This is representative of behavior observed in many studies (e.g., Zielke et al., 2010; Klinger et al., 2011; Zielke, 2018). Taken together, the mean slip of 1 m and the mean recurrence interval of 1,000 years shared by each of the recurrence interval distributions yields a mean slip rate of 1 mm/yr. This rate is fairly typical for intraplate faults studied by paleoseismologists, and also allows for easy normalization so that the results of this study can be generalized to faults with different parameters.

The choice of the lognormal distribution is for convenience, simplicity, and flexibility: it is a common, well-known distribution and–should one be interested–can be easily given different shape and scale values to modify the $CV$ or change the mean slip rate in the modeling code used in this paper.

However, it is not necessarily the most accurate representation of earthquake slip variability. Biasi and Weldon (2006) compiled field measurements of surface ruptures from 13 earthquakes. The resulting distribution (Figure S1) has some significant differences with the lognormal distribution used here, though the $CV$ of 0.67 is quite close to the value used here. To test the sensitivity of the results given in this paper to the choice of slip distribution, the numerical simulations in presented in this work were run with the only change being the use of the empirical slip distribution from Biasi and Weldon (2006), and the results are given in the supplemental material (Figure S2, Table S1). Though there is more discussion in the supplements, the results are essentially identical to those presented below. As an additional experiment,

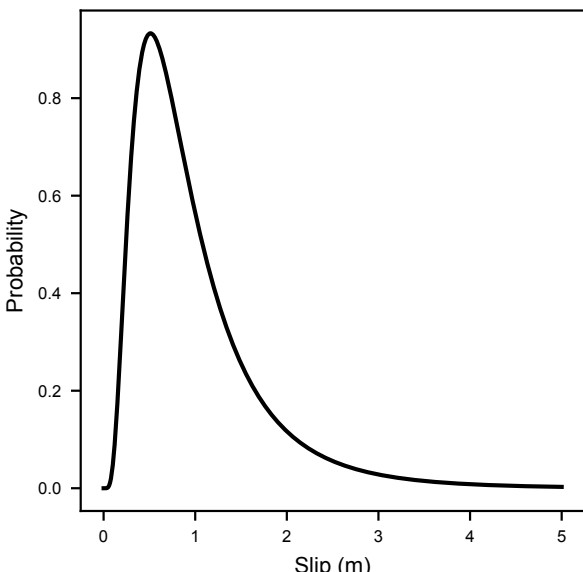

**Figure 3.** Earthquake slip distribution.

the numerical simulations have been run using an invariant per-event displacement of 1 m. Though this is not a realistic scenario, it allows for a deconvolution of the effects of earthquake time stochasticity and earthquake displacement stochasticity. The results are shown in Figure S3 and Table S2, and discussed in the supplemental materials; there are noticeable differences in the results, but they are quite small and do not call into question the results and conclusions presented below.

## 2.2 Stochastic displacement histories

For each of the earthquake recurrence distributions, a 2 million year long time series of cumulative displacements is calculated, and then slip rates are estimated over time windows of different lengths.

The construction of the displacement histories is straightforward. From each recurrence distribution, a little over 2,000 samples are drawn randomly. Then, these are combined with an equal number of displacement samples drawn randomly from the earthquake slip distribution. Finally, a cumulative displacement history is created for each series from a cumulative sum of both the recurrence interval samples (producing an earthquake time series) and displacement samples (producing a cumulative slip history). Years with no earthquakes are represented as having no increase in cumulative displacement. Then, the series is trimmed at year=2,000,000; it is initially made longer because the stochastic nature of the sample sets means that 2,000 earthquakes may not always reach 2,000,000 years.

The displacement histories in Figure 4 clearly show that given the stochastic nature of the samples, the cumulative displacements can diverge greatly from the mean. The magnitude of this divergence appears to be related to the *CV* of the recurrence interval distributions: The clustered series (*CV* = 2) has by far the most divergence, both unclustered series (lognormal and exponential with *CV* = 1) behave qualitatively similarly, and the periodic series (*CV* = 0.5) tracks most closely with the mean. The divergences from the mean are driven by successive closely-spaced earthquakes, perhaps with high displacements, or by long durations of quiescence. The clustered series in particular shows a pattern of many closely-spaced events (clusters) leading to a much higher than average displacement accumulation rate, followed by very long episodes of dormancy in which regression to the mean occurs. From visual inspection, the dormant episodes appear to be composed of single or dual exceptionally long inter-event times. This of course is reflected in the great asymmetry of this distribution (Figure 1), with the very short mode and 'fat' right tail.

Please note that in the construction of the cumulative displacement histories, all samples are independent. This means that the duration of any recurrence interval does not depend on the duration of the previous or subsequent interval (in other words, there is no autocorrelation in these series); the same applies to the displacement samples. It is currently unknown to what degree autocorrelation exists in real earthquake time and displacement series, or how much correlation is present between recurrence intervals and subsequent displacements. Autocorrelation in recurrence interval sequences is essentially unstudied, though on the basis of preliminary, unreviewed analysis (Styron et al., 2017), I suspect that it is as important as *CV*.

Furthermore, the magnitude of displacement is independent of the corresponding recurrence interval. The framework of elastic rebound theory in its most basic form should predict some correspondence between inter-event (loading) duration and slip magnitude, and this is included (implicitly or explicitly) in oscillator models incorporating complete stress or strain release in each earthquake (e.g., Matthews et al., 2002; DiCaprio et al., 2008) or in any model where coseismic friction drops to zero, as this is functionally equivalent (because $f^a = \tau_s^a / \tau_n^a$, where $f^a$, $\tau_s^a$ and $\tau_n^a$ are respectively friction at rupture arrest, shear stress at rupture arrest, and effective normal stress at rupture arrest, zero friction implies zero shear stress, or complete stress drop). Given a reasonably constant loading rate, complete shear stress or strain release implies some proportionality between the loading time and displacement. Nonetheless, this correspondence is not found in the more extensive paleoseismic datasets, such as those by Benedetti et al. (2013) (or the correlation may be negative as found by Weldon et al. (2004)), but the number of paleoseismic datasets of sufficient size and cquality to identify these effects with statistical significance are few indeed.

Because this modeling strategy involves sampling independence, it is essentially a neutral model. If any correlation structure exists in the sample sets, it will affect the displace-

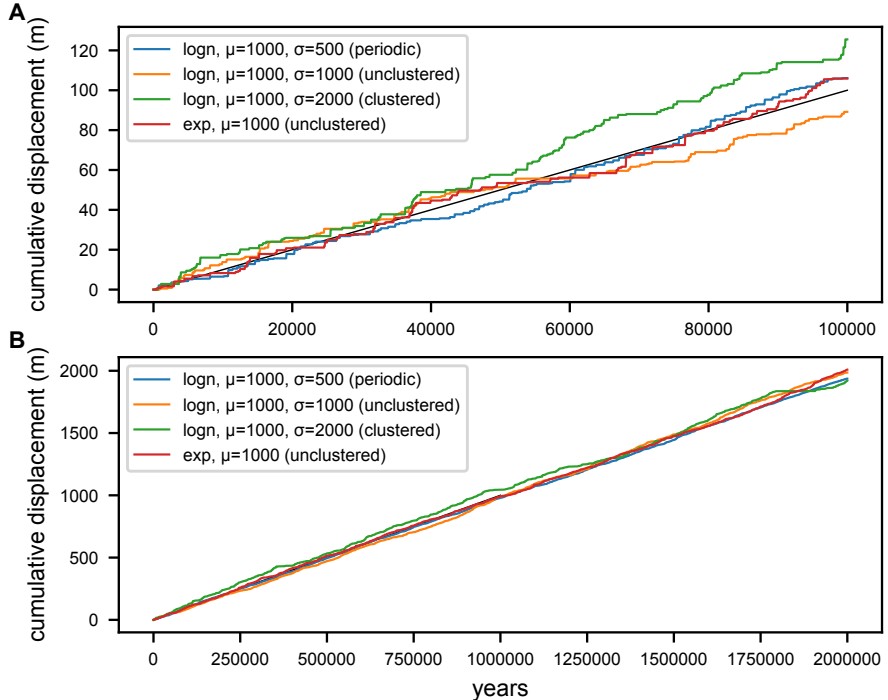

**Figure 4.** Simulated displacement histories for each of the recurrence distributions, and the 'true' mean line at 1 mm/yr in black. *A*: The first hundred thousand years. *B*: The entire 2 million years. The histories are the same in both plots.

ment histories in predictable ways. Negative autocorrelation in the sample sets, meaning that a long interval (or slip distance) is followed by a short interval (or slip distance) and vice versa, will cause a more rapid regression to the mean slip rate line, and decrease the scatter in the slip rate estimates. A positive correlation between recurrence (loading) intervals and slip magnitudes will have the same effect. Conversely, positive autocorrelation in either of the sample sets, or negative correlation between the recurrence intervals and slip magnitudes, will lead to slower regression to the mean line and therefore an increase in the scatter of the slip rate estimates.

## 2.3 Slip rate calculations

The uncertainty in the estimated slip rates due to earthquake cycle variability is estimated by taking a function, $\hat{R}$, that calculates the mean slip rate within a time window $t$, and sliding it along the displacement series. $\hat{R}$ is calculated simply as

$$\hat{R}(D_0, D_1, t) = \frac{D_1 - D_0}{t} \, , \qquad (1)$$

where $D_0$ is the cumulative displacement at the beginning of the time window, $D_1$ is the cumulative displacement at the end of the time window, and $t$ is the length of the time window. The ˆ symbol signifies an estimate rather than the true value $R$. This slip rate estimation method is intended to represent a neotectonic-style slip rate estimate in which the number of earthquakes that have contributed to the observed deformation is unknown, as are the durations of the open intervals that bound the time window (one of which preceeds deposition of the marker unit, and one is the time since the most recent earthquake and the measurement time). By sliding $\hat{R}$ over the displacement series, a set of many samples of $\hat{R}$ is generated, so that we may analyze the distribution. The number of samples is $n = N - t + 1$, where $N$ is the length of the total series (2,000,000 in this study).

A major goal of this study is to provide an answer to the question, *How long should slip rates be measured over in order to estimate a meaningful rate?* This question will be answered by looking at the distribution in $\hat{R}$ as a function of $t$. Fifty values of $t$ from 500 years to 100,000 years, logarithmically spaced, are used. Note that given $\mu$ of 1000 years, this translates to 0.5–100 mean numbers of earthquakes in the window.

The results of these calculations are shown in Figure 5 for up to 60 mean earthquake cycles. It is clear that the total variability in the estimated slip rates is initially quite high when $t$ is short (<10,000 years or ~10 earthquakes). Particularly when $t$ is <5,000 years, the maximum rates are a factor of 3 or more greater than the true rate $R$, but the median rates are lower than $R$—this means it is more likely that fewer earthquakes are captured in the time window than naively expected given the mean recurrence, and that the time contained in the open intervals is a substantial fraction of the

total time window. As the median is lower than $R$, most measurements over these short timescales will underestimate the mean rate, although not necessarily by much.

With longer $t$, between 10,000–20,000 years (or 10–20 earthquakes) the variation in the slip rate estimates stabilize to within $\pm$ 100% of the mean (Figure 5) for all distributions, though this happens most quickly in the periodic distribution, and most slowly in the clustered distribution. In fact, the only exception here is that the lower bound of the clustered distribution can stay at zero for more than 60 mean earthquake cycles. It is highly unlikely that any given recurrence interval will be this long, but given thousands of earthquakes over millions of years, the chance of such an event occurring at least once is far more likely. For rate estimates longer than several tens of mean earthquake cycles, the variation decreases very slowly but progressively with increasing window length.

Note that with a measurement time exceeding 5–10 mean earthquake cycles, the standard error $(n/\sqrt{(\sigma_D^2 + \sigma_r^2)}$, where $\sigma_D$ is the standard deviation of displacement, and $\sigma_r$ is the standard deviation of the recurrence interval) is a reasonable approximation for the standard deviation of the variability in the slip rates due to earthquake-cycle stochasticity, and shows broadly similar decreasing variation with increasing earthquake cycles. However, the standard error is symmetrical, though the variability displayed here is asymmetrical due to the asymmetry of the recurrence interval and displacement distributions.

### 2.3.1 Normalizing to different slip rates and earthquake offsets

The distributions in this study were chosen to have $\mu = 1$ (kyr, m) in order to make the mean slip rate $R = 1$ mm/yr, and therefore to make all results easy to generalize to different systems with different real rates. This normalization requires some values for the mean per-event displacement $\bar{D}$ and the slip rate $R$, yielding a normalization factor (or coefficient) that can be applied to the time values shown in as the $x$-axis in Figure 5:

$$NF = \frac{\bar{D}}{R} \qquad (2)$$

$NF$ is also equal to the mean recurrence interval $\mu$ given suitable unit transformations (though the recurrence interval may not be known *a priori*). For example, a fault with a slip rate of 5 mm/yr but a per-event mean slip of 1 m has a normalization factor of 0.2, meaning that earthquakes are 5 times as frequent on this fault as the simulated fault, so the time window required for the rates to stabilize is 0.2 times the simulated fault. For a fault with $R = 1$ mm/yr and $\bar{D} = 2.5$ m, $NF = 2.5$, then the mean recurrence interval $\mu$ is 2.5 times as long as in these simulations, so $NF = 2.5$ and the timescales for rate stabilization will be lengthened that much.

This normalization will obviously be more accurate if $\bar{D}$ and $R$ are independently (and accurately) known or can be obtained from other information. $\bar{D}$ may be estimated paleoseismologically or through the application of scaling relationships between fault length and offset (Wells and Coppersmith, 1994; Leonard, 2010). The accuracy of $\hat{R}$ is discussed below, but suffice it for now to state that for more than ~10 earthquakes, $\hat{R}$ should be acceptable.

## 3 Discussion

### 3.1 Interpreting measured rates

The most pragmatic motivation for this study is to understand how much *epistemic* uncertainty in a slip rate measurement results from the *aleatoric* (or natural) variability in earthquake recurrence. However, the previous results have focused on describing the natural variability, and how much a measured rate may deviate from the 'true' secular rate, i.e. $\hat{R}/R$. In these methods and results, there is no epistemic uncertainty because all quantities are known perfectly. Of course, in a real slip rate study, the measured value is known, but the true value is not. The epistemic uncertainty then is present, and can be quantified here by evaluating the true rate $R$ relative to the measured rates $\hat{R}$, so that the distribution of $R/\hat{R}$ at a given $t$ represents the epistemic uncertainty distribution about the measured value.

The epistemic uncertainty relative to the measured rate is shown in Figure 6 for all distributions for the first 40,000 years (or 40 earthquakes), represented by the 5, 25, median, 75 and 95 percentiles, and numerical results are given in Table 1. Several things are clear in these plots:

First, the variance in the distributions is quite large for the first several thousand years (or several mean earthquake cycles), but becomes much more compact after ~15 mean earthquake cycles, as with the slip rate estimates in Figure 5. The right tails (or upper bounds in Figure 6) in fact are infinite (or undefined) for the first few earthquake cycles, because in some fraction of the simulations, $\hat{R}$ is zero.

Second, the distributions are asymmetrical, especially the 5–95% interval. The 95th percentile is generally several times as far from the measured value as the 5th percentile, meaning that the true value of the slip rate may be much greater than the measured rate but not a commensurately small fraction of the measured rate.

Third, the median rate before convergence at 5 $\mu$ is greater than the true rate, meaning that in most cases, very short-term slip rate measurements will underestimate the true slip rate; this is a systematic bias. This is a particular concern for paleoseismological slip rate estimates, where there are rarely more than 5 events in any given trench, though this bias will be decreased if slip rate measurements are made from closed intervals only. However, the median is less than 2 times $\hat{R}$ following just 2–3 events, so the systematic bias is unlikely

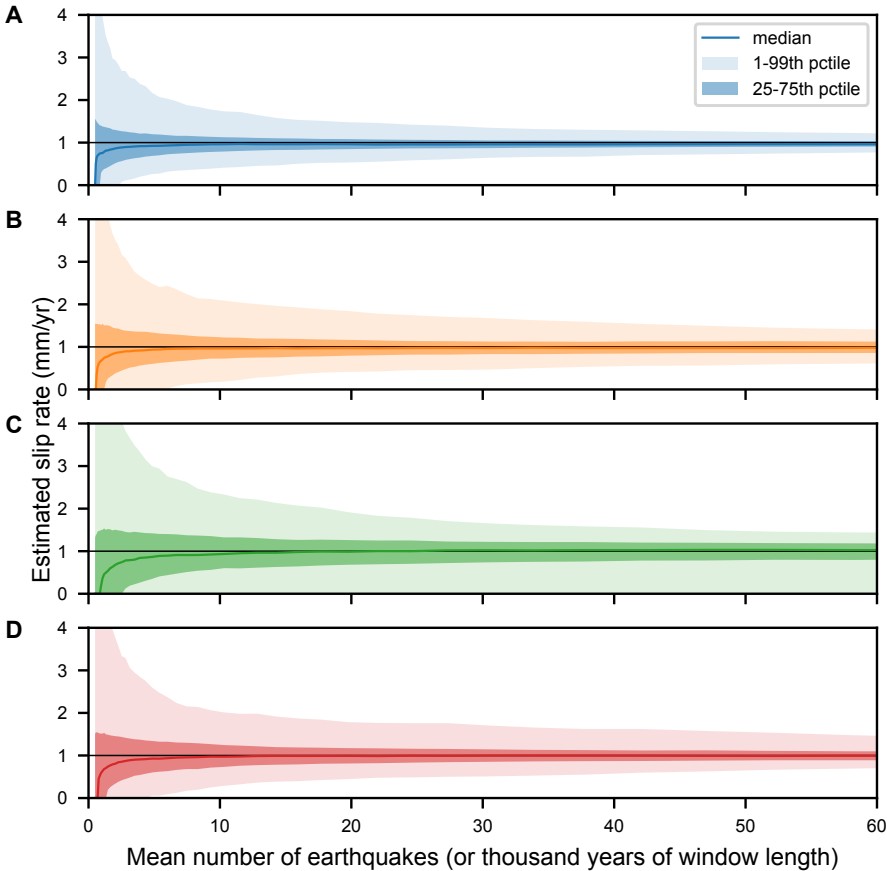

**Figure 5.** Envelopes of estimated slip rates as a function of the mean number of earthquakes (or thousands of years) over which the slip rate was estimated. All slip rates have a true value of 1 mm/yr. *A*: periodic distribution. *B*: unclustered lognormal distribution. *C*: Clustered lognormal distribution. *C*: Unclustered exponential distribution.

to be much greater than the measurement uncertainties on the age or offset of the events.

### 3.2 Evaluating slip rate changes

It is of both theoretical and practical interest to be able to evaluate whether fault slip rates may have changed over some time period, or between multiple sets of measurements. From a theoretical perspective, understanding under what conditions fault slip rates change can lead to much insight into fault processes such as growth (e.g., Roberts et al., 2002) and interaction (e.g., Wallace, 1987; Dolan et al., 2007). Practically, if an older (or longer-term) slip rate is quite different from the contemporary rate, then its inclusion in a seismic hazard model may lead to inaccurate hazard estimation.

First, a necessary definition: A slip rate change in this discussion means a real change in $R$, not a change in the estimate $\hat{R}$, which leads to a change in the distribution of earthquake recurrence and/or displacement parameters with time (in statistical terminology, the recurrence and displacement distributions are then *non-stationary*). This sort of change may be

associated with secular changes in fault loading, stemming from changing stress or strain boundary conditions.

Discerning a real slip rate change, rather than a change in $\hat{R}$ due to natural variability, requires consideration of the lengths of time over which the different slip rate measurements were made and the associated uncertainty. If the distributions defined by two estimates $\hat{R}_1$ and $\hat{R}_2$ and their empirical distributions (reflecting the number of earthquakes as well as the $CV$s of the underlying earthquake recurrence distributions are known, then the null hypothesis that the two slip rates are drawn from the same stationary distribution can be tested with a Kolmogorov-Smirnov test.

However, it is unlikely that the values for the recurrence distributions and the number of earthquakes that have transpired are sufficently known to make the calculation, unless the fault has received in-depth paleoseismic and neotectonic study. As formal hypothesis testing may not be possible given typical slip rate datasets, an informal way of gauging the likelihood of a slip rate change is to crudely estimate ('guesstimate') the number of possible earthquakes and the recurrence

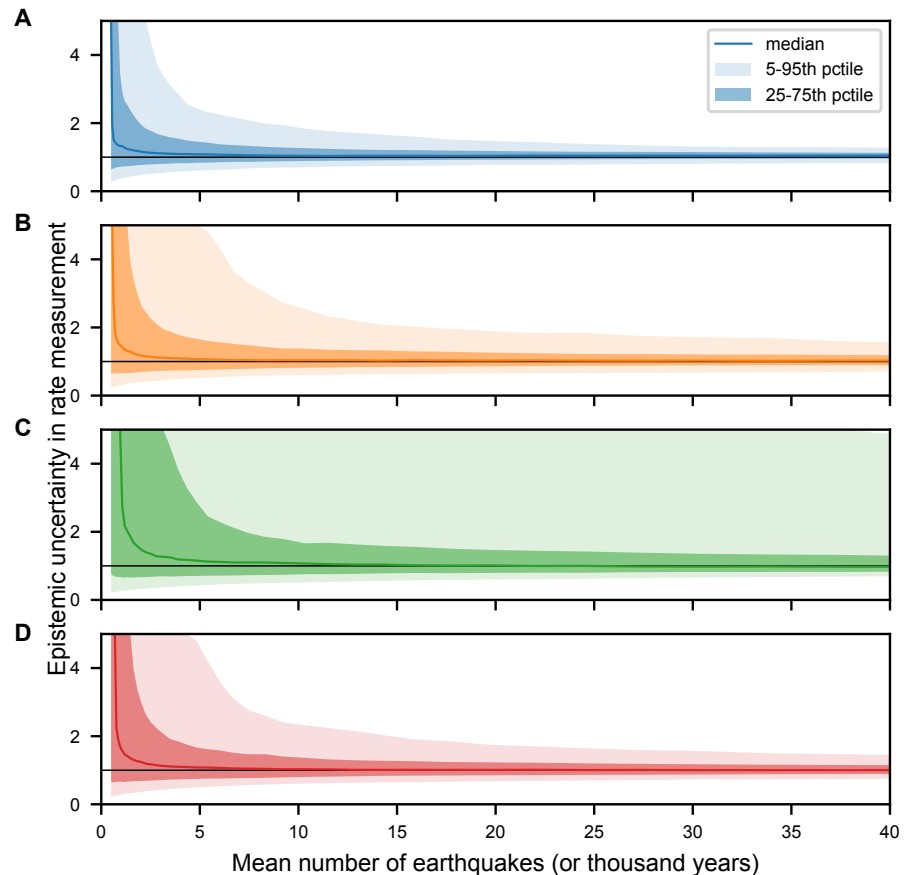

**Figure 6.** Epistemic uncertainty relative to the measured rate for each of the recurrence distributions, as a function of the mean number of earthquakes (or thousands of years) over which the slip rates were measured. *A*: periodic distribution. *B*: unclustered lognormal distribution. *C*: Clustered lognormal distribution. *C*: Unclustered exponential distribution.

distribution, and then use the closest values in Table 1 with propagated measurement uncertainty to evaluate the amount of overlap between the two slip rate estimates. If the overlap between the distributions is a small fraction of the total range of the distributions, then it is likely that a real slip rate change occurred. This is clearly not appropriate for a real hypothesis test, but it may aid researchers in developing ideas or intuition about the behavior of a given fault.

## 4   Conclusions

This work seeks to evaluate the effect of natural (aleatoric) variability in earthquake recurrence intervals on slip rate measurements. The study simulates cumulative displacement during 2,000,000 earthquakes for a faults with stationary long-term slip rates of 1 mm/yr and several different distributions for earthquake recurrence, and then estimates the variation in estimated slip rates over shorter time windows similar to those measured in paleoseismological and neotectonic studies. The results display several characteristics that

are of importance to fault geologists seeking to estimate slip rates on faults, or compare rates from different measurement time windows or techniques:

1. The variability in slip rates calculated over time windows less than 5 mean earthquake cycles is very large, but begins to stabilize following ~10–20 earthquakes.

2. The most important factor in controlling the variability in slip rate estimates is the coefficient of variation (*CV*); the different distributions themselves are relatively unimportant. Faults with *periodic* earthquakes, with *CV* < 1, have less initial variability, and stabilize rapidly. Faults with *unclustered* earthquakes, with *CV* = 1, have more variability and stabilize more slowly. Faults with *clustered* earthquakes, with *CV* > 1, have a great amount of initial variability and require a large number of earthquakes to stabilize.

3. The epistemic uncertainties around a measured slip rate are similarly large initially and then decrease with time. These uncertainties are initially biased, such that the

| distribution | $t$ | 5% | 25% | 50% | 75% | 95% |
|---|---|---|---|---|---|---|
| lognormal (CV=0.5) | 2531 | 0.51 | 0.79 | 1.13 | 1.77 | 4.37 |
| | 4843 | 0.61 | 0.84 | 1.08 | 1.46 | 2.44 |
| | 10,323 | 0.7 | 0.89 | 1.04 | 1.27 | 1.84 |
| | 42,103 | 0.83 | 0.96 | 1.04 | 1.12 | 1.27 |
| lognormal (CV=1) | 2531 | 0.42 | 0.71 | 1.14 | 2.25 | ∞ |
| | 4843 | 0.51 | 0.75 | 1.07 | 1.67 | 5.62 |
| | 10,323 | 0.6 | 0.82 | 1.04 | 1.38 | 2.55 |
| | 42,103 | 0.72 | 0.89 | 1.03 | 1.18 | 1.55 |
| lognormal (CV=2) | 2531 | 0.35 | 0.68 | 1.35 | ∞ | ∞ |
| | 4843 | 0.43 | 0.7 | 1.16 | 2.86 | ∞ |
| | 10,323 | 0.51 | 0.75 | 1.07 | 1.66 | ∞ |
| | 42,103 | 0.7 | 0.83 | 0.98 | 1.28 | 4.75 |
| exponential (CV=1) | 2531 | 0.4 | 0.7 | 1.17 | 2.39 | ∞ |
| | 4843 | 0.49 | 0.75 | 1.08 | 1.67 | 4.79 |
| | 10,323 | 0.61 | 0.8 | 1.03 | 1.37 | 2.32 |
| | 42,103 | 0.76 | 0.89 | 1 | 1.15 | 1.43 |

**Table 1.** Epistemic uncertainty tabl, showing the percentiles for the slip-rate variability (in mm yr$^{-1}$) at each time $t$ (years). The long-term mean slip rate is 1 mm yr$^{-1}$.

measured slip rates typically underestimate the true slip rate for <5 mean earthquake cycles, but this fades with time. However, the uncertainties remain asymmetric, with a strong right skew.

*Code availability.* All code is available at https://github.com/cossatot/eq-slip-rate-variability-paper with an MIT license.

*Competing interests.* The author declares no competing interests.

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
