# Peer review of "The impact of earthquake cycle variability on neotectonic and paleoseismic slip rate estimates"

_Solid Earth, 2018_

## Referee Comment (RC1) · Anonymous Referee #1 · 28 Jun 2018

A well written short and concise paper. Not much to change before it is ready for publication. Below are a few comments by page/line. hope it helps.

P1L1: I suggest to use "aleatoric variability" and "epistemic uncertainty". That way there is only one "variability" and one "uncertainty", which makes the language more clear. Please modify throughout the manuscript.

P1L9: I believe the mathematicians call it just CV and not COV. Maybe a good idea to stick to the prior naming convention.

P1L10: This statement "... is quite high"... is a bit too vague. Better add numbers (COV values) here as well.

P1L23: Putting the "e.g." at the end of a list of references seems unusual. Is this an

accepted format for this journal? Please check and modify in necessary.

P1L1: The connection to locking depth should be explained a bit more. Good to also provide a reference here.

P3L29: I find it troublesome to talk about periodic/regular occurrence just because CV is smaller than 1. That would be correct for CV = 0, as you also pointed out. Depending on CV value between 0 and 1, it might be better to talk about quasi-periodic, or quasi-random behavior.

P4L26: Using this distribution seems plausible. It would however be really interesting to see other distributions explored –if possible, that would be a great addition to make the manuscript more complete.

P5L17: Here you describe qualitatively how more or less closely the different distributions align with the mean slip rate. While doing this qualitatively is ok to first order, I suggest that you go one step further and compute some form of misfit function i.e., residual (simplest a L1 or L2 norm).

P6L2: stress drop doesn't need to be "complete" –just has to be "the same" each time to get to the outcome you describe here. Maybe better rephrase accordingly.
* * *

---

## Short Comment (SC1) · 17 Jul 2018

A well written short and concise paper. Not much to change before it is ready for publication. Below are a few comments by page/line. hope it helps
* * *

---

## Author Comment (AC1) · 18 Jul 2018

Dear John,

Many thanks for your interest in my paper. Though you have indicated in your comment that you have a list of small suggested line edits, I don't see that they were successfully attached. I would happily consider them if you repost them.

Thanks, Richard

---

## Referee Comment (RC2) · M. Oskin (Referee) · 27 Jul 2018

This paper presents a useful thought experiment on the impact of earthquake cycle variability on measured slip rates, and concludes that the convergence on the expected value is a function of the coefficient of variation. Overall this is a sensible conclusion. Underpinning this analysis are four assumed variants of earthquake recurrence, and a function to express the variability of slip per event. I would like to see the effect of COV isolated from the slip per event distribution (use 1m slip for every event). I would also like to a see a more quantitative comparison of COV and a convergence on the mean to back up the assertion that COV of the distribution is more important than the distribution itself. The paper would be improved by a more quantitative, empirical basis and discussion of physical processes that may drive such recurrence behavior.

[Figure]

There is a literature of ideas to draw upon, such as post-seismic fault reloading (Kenner and Simons, 2005), earthquake super cycles (Sieh et al., 2008; Weldon et al., 2004), isolated versus fault-network behavior (Berryman et al., 2012). Some of these ideas are discussed briefly but need more explanation. Likewise one could examine actual earthquake slip distributions (not landform offsets of historic events, which convolve landscape processes with tectonic slip) to develop an empirical basis for the slip function. Some of the scatter in slip distributions is likely due to underreported measurement uncertainty (Gold et al., 2013), and thus the cancellation of this error over multiple earthquakes should let cumulative slip converge more quickly than may be predicted from the author's model.

Line-by-line comments:

Page 1, line 4. The open interval problem is well known and attempts to quantify it do exist on case-by-case basis.

Page 1, line 13. It seems odd to characterize uncertainty due to a random distribution as epistemic. Isn't this unreported aleatory uncertainty?

Page 1, line 20: Why is marker in quotes?

Page 2, line 5. afterslip and creep also contribute.

Page 2, line 10-11. Awkward sentence. Break into two.

Page 2, line 13 and other citations: persistent use of 'e.g' after citing only one or two articles is poor form and makes this reader think that the author has not adequately explored the literature.

Page 3, line 11. This is not the correct definition of an exponential / poisson distribution. There is no prescribed number of events, only a prescribed time-independent probability. It is also worth noting that this is physically unrealistic at short time intervals because it violates elastic rebound.

Page 3, line 24. It would be useful to briefly discuss how shape and scale affect distributions generally. Shape governs the how tailed and is dimensionless; scale determines the spread of the distribution and is dimensioned (in years for this case).

Page 4, line 6. Pareto distribution is another, simpler distribution needing only shape and scale to describe COV > 1

Page 4, line 15. Akciz et al (2012) revised Grant and Sieh (1994) and found much more periodic behavior.

Page 4, line 28. The author should consider non-dimensionalizing the results of this study to facilitate more general use of its results. Instead of mean slip of 1m, one would refer to non-dimensional slip of 1 and multiply by average slip per event to scale the results. This is effectively what the author describes already, though without formal non-dimensionalization.

Page 5, line 15. The statement 'appears to be related to COV' is disappointing. Given that this paper is entirely simulation, the author should be able to make a quantitative comparison of slip-rate variance to COV.

Page 5, line 24. A 'fat' or heavy tailed distribution would not have a defined variance, nor COV. The author should refer to this as a long tail.

Page 5, line 29. This problem has been studied (Weldon et al., 2004; Sieh et al., 2008). The Sumatran subduction zone work is particularly relevant and completely overlooked here.

Page 6, line 1. Zero friction at rupture arrest is very unrealistic, and not a prerequisite for characteristic behavior.

---

## Referee Comment (RC3) · Anonymous Referee #3 · 1 Nov 2018

Review, R. Styron, "The impact of earthquake cycle variability on neotectonic and paleoseismic slip rate estimates."

Uncertainty in estimates of earthquake recurrence and fault slip rate are important parameters, pursued by conscientious investigators of seismic hazard. The author perceives a lack of statistical support, and offers in this paper "insights" in variously tones of "friend of the practitioner" and "trust me, I'm the numerate one here". Neither is convincing.

A couple of omissions in this paper are particularly striking. First, how do we have a paper addressed to "variability" in fault slip rate, addressing particularly the problem of small samples, without mentioning the methods of estimation for censored samples? There is an extensive statistical literature to estimate parameters and uncertainties and numerous recent papers applying it in paleoseismic contexts. This literature provides real quantitative methods to deal with the open intervals, long or short, that affect the geologist's estimate of fault slip rate and recurrence estimation. These are real equations, with real uncertainties. One would look in vain in this paper for anything of similar substance. Second, pages of this paper could be replaced (and improved) by a presentation and discussion of the properties of the standard error. E.g., given an estimate a sample-based estimate of the mean, how far might the population (or true) mean be from the estimate? S.E. is estimated by the sample standard deviation divided by the square root of the number of samples. So, of course, estimates from small samples from a fuzzy log normal converge more slowly than from a well-defined (quasi-periodic) lognormal. Instead of a small equation (SE=s/sqrt(n)), our paper back-calculates the result using 2 million years of samples, and presents the results like a new discovery. And again, with little by way of meaningful uncertainties (e.g., p1, lines 11-13, 14-16).

A few particulars

1 L9,10: We read that the most important parameter is the coefficient of variation. First, this equation is the arithmetic coefficient of variation, and not the CV for a lognormal distribution. The CV of a lognormal does not depend on the mean. We could stop here, but a central flaw in the paper is exposed – nothing in this paper addresses how to obtain this most important parameter. If attempted, the essential emptiness of a 2,000,000 year sample would emerge. No real data set in paleoseismology resolves the mean and standard deviation to better than maybe 50%. Typical sites do well to resolve it to a factor of 2. P.3, line 22-23 reflect this reality.

It is not obvious that the author has material experience words "aleatoric" and "epistemic". Line 1, "aleatoric uncertainty" is a contradiction in terms. Bird, Zechar and Frankel all know better than to use the method the author alleges in lines 21 and 22 to arrive at epistemic uncertainty in slip rate. They would more likely consider the allegation a misreading of their work. I could multiply examples. More broadly, the lack of care in writing makes one wonder how to understand this paper. p2, L5: A perturbation in slip rate would mean it was slipping at rate X, then changes to Y. p.2, L14-17 have careful paleoseismologists doing reasonable things in one sentence, then imply they would make plainly rookie mistakes in the next. From here these read like inexperienced generalizations.

p.4, L15-23:  The descriptions of the lognormal variables here give one pause.  First, log-normal parameters do not have units.  Second, the mean recurrence interval is not the location parameter of a log normal.  This is just wrong.  Third, if one uses the CV equation for the lognormal distribution (e.g., [https://en.wikipedia.org/wiki/Log-normal_distribution#Arithmetic_coefficient_of_variation)](https://en.wikipedia.org/wiki/Log-normal_distribution#Arithmetic_coefficient_of_variation),  the CV will not match the COV alleged here.   Given that the study depends on these distributions, we can't really use subsequent conclusions.

p.6, L20.  If the number of samples is really $n = N - t + 1$, the samples are correlated by virtue of the overlap in the windows.  No accounting has been made of the correlation structure.

p.6, L26:  Starts a narrative of the consequences of the standard error, as though the standard error was never invented.  The fuzzy, back-of-the-envelope estimates start to get thick here.  Real uncertainty estimates would serve better.

p.7, L1-3:  Two observations:  First, as written, the practicing geologist is being asked to believe that 60 earthquake cycles have passed with zero displacement.  I can guess what was intended, but should not have to.  Second, what probability is associated with this 60-cycle thing?  I ask because practicing hazard geologists have to make estimates, and give weights to extreme events.  What is the probability of 60 cycles, a CV of 2.0, …?  Hard to imagine that the author has thought much about what these results would mean or how to use them if they were true.

---

## Author Comment (AC2) · 28 Nov 2018

The comment was uploaded in the form of a supplement:
https://www.solid-earth-discuss.net/se-2018-40/se-2018-40-AC2-supplement.pdf

---

## Author Comment (AC3) · 28 Nov 2018

The comment was uploaded in the form of a supplement:
https://www.solid-earth-discuss.net/se-2018-40/se-2018-40-AC3-supplement.pdf

---

## Author Response (AR1)

**Response to Reviewers, "The impact of earthquake cycle variability on neotectonic and paleoseismic slip rate estimates"**

**Richard Styron**

*Reviewer comments denoted as [Ax], [Ox], or [Bx], and refer to Reviewer 1, Reviewer 2 (Michael Oskin), and Reviewer 3, respectively.*

**Review by Anonymous Reviewer #1**

[A1] P1L1: I suggest to use "aleatoric variability" and "epistemic uncertainty". That way there is only one "variability" and one "uncertainty", which makes the language more clear. Please modify throughout the manuscript.

This is a fine suggestion.

*Changes: modified terminology.*

[A2] P1L9: I believe the mathematicians call it just CV and not COV. Maybe a good idea to stick to the prior naming convention.

Also a good suggestion.

*Changes: modified acronym.*

[A3] P1L10: This statement " ... is quite high" ... is a bit too vague. Better add numbers (COV values) here as well.

I modified the sentence in question to state that the rates may vary by a factor of 3 or on short (<5,000 year) timescales.

*Changes: Numbers added.*

[A4] P1L23: Putting the "e.g." at the end of a list of references seems unusual. Is this an accepted format for this journal? Please check and modify in necessary.

It was a LaTeX error on my part, and has been fixed.

*Changes: LaTeX fix.*

> [A5] P1L1: The connection to locking depth should be explained a bit
> more.
> Good to also provide a reference here.

I added a phrase stating that the width of the zone affected by earthquake-cycle strains is a function of the fault's locking depth, and added refs to Savage and Burford (1973), the classic locked-fault-above-creeping-fault reference, and Hetland and Hager (2006), which demonstrates that this process can instead be the result of post-seismic relaxation.

*Changes: explanation and references added.*

> [A6] P3L29: I find it troublesome to talk about periodic/regular occur-
> rence just because CV is smaller than 1. That would be correct for CV =
> 0, as you also pointed out. Depending on CV value between 0 and 1, it
> might be better to talk about quasi-periodic, or quasi- random behavior.

The referenced sentence states that in this paper, I'm not using the word 'periodic' to mean perfectly periodic behavior. In any case I added a wiggle phrase stating that I mean 'quasi-periodic' but I won't change the word through the whole paper because it would decrease readability a bit.

*Changes: terminology explained.*

> [A7] P4L26: Using this distribution seems plausible. It would however
> be really interesting to see other distributions explored –if possible, that
> would be a great addition to make the manuscript more complete.

I have added another numerical experiment using an empirical slip distribution; see the response to [06a].

*Changes: numerical experiment w/ alternate slip distribution added.*

> [A8] P5L17: Here you describe qualitatively how more or less closely
> the different distributions align with the mean slip rate. While doing this
> qualitatively is ok to first order, I suggest that you go one step further
> and compute some form of misfit function i.e., residual (simplest a L1
> or L2 norm).

An L-norm of any sort isn't appropriate because the numbers here aren't pairwise (observation, model) data. Instead, there is a single 'true' value at any time $t$, which is invariably 1, and then hundreds to thousands of simulated values at each time $t$.

Instead, because the true value is 1, all of the results shown in Figure 5 are quantified percentiles of misfit. While I could calculate the CV as suggested in comment [A3], this is pretty reductive and the numbers at any time $t$ can be retrieved from Figure 5.

*Changes: None.*

> [A9] P6L2: stress drop doesn't need to be "complete" –just has to be "the same" each time to get to the outcome you describe here. Maybe better rephrase accordingly.

No, this isn't true. The outcome that I describe is a *correlation* between loading time and displacement, not an *invariance* of either loading time or displacement. Having 'the same' stress drop doesn't predict correlated loading time/recurrence intervals and displacement.

It's possible that this paragraph was too confusingly written for it to be easily interpretable. I have made some minor changes to the sentence structure for clarification

*Changes: Paragraph edits.*

**Review by Michael Oskin**

> [O1] This paper presents a useful thought experiment on the impact of earthquake cycle variability on measured slip rates, and concludes that the convergence on the expected value is a function of the coefficient of variation. Overall this is a sensible conclusion. Underpinning this analysis are four assumed variants of earthquake recurrence, and a function to express the variability of slip per event.

> [O2] I would like to see the effect of COV isolated from the slip per event distribution (use 1m slip for every event).

I performed this experiment; the figure with the slip rate results is shown below for comparison with Figure 5 in the paper, and are included in a new document in the supplemental material. The most relevant figure is also shown here (Figure 1)

The differences between the results of this experiment (fixed per-event displacement of 1 m) and of the numerical experiment performed in the manuscript are basically that these results are less smooth, but the total variance at any point in time (*x*-axis) is less. This is simply due to removing the stochasticity from one of the two variables in the system. The relative spread in the data and convergence rates are unchanged. This is to be expected as the variability in the per-event displacement is the same for all recurrence distributions, so even though it is a random variable in the simulations, it is not an experimental variable.

What I find the most interesting about this experiment is that the fluctuations in the estimated slip rates show very clearly the mean earthquake cycles once the noise from the per-event displacements has been removed. It is clear that these are kind of damped or averaged out after ~7 earthquake cycles. Nonetheless, though this is a cool pattern to see, I don't think it adds enough insight to be worth including in the manuscript. I have added the figure and table that show the slip rate variation through time to a new supplemental document, accompanied by a brief discussion.

*Changes: New experiment added to new supplemental materials, and brief discussion in the main manuscript.*

[Figure]

Figure 1: Slip rates with no displacement variability

[O3] I would also like to a see a more quantitative comparison of COV and a convergence on the mean to back up the assertion that COV of the distribution is more important than the distribution itself.

This comment is addressed in the response to [O18] below.

[O4] The paper would be improved by a more quantitative, empirical basis and discussion of physical processes that may drive such recurrence behavior.

I don't understand how the work could be more quantitative–it's a purely numerical study.

As for making it more empirical, there are several ways to do this:

1. *Using empirical distributions for earthquake recurrence and slip.* There isn't a lot of consensus on empirical (non-parametric) earthquake recurrence distributions; most of the community prefers parametric distributions such as the lognormal, Weibull or Brownian Passage Time distributions. As discussed in the paper by Matthews et al. (2002) that is the most prominent introduction to the Brownian Passage Time distribution, given the very small number of samples for earthquake recurrence that we will have for a given section of faulting, it is impossible to discriminate between these distributions, so going with the lognormal (as I have done in this study) is justifiable on empirical grounds as well as practical ones (it is familiar, implemented in many programming environments, and easy to manipulate). With regards to empirical slip rate distributions, I have added an experiment that uses one, which is explained in more detail in response to comment [O6a] below.

2. *Going through the literature and evaluating studies that claim that slip rates have changed (or have not changed) in light of the work presented here.* I considered this, and in fact a major motivation for me to begin the analysis was my skepticism over some recent literature claiming major slip rate changes over relatively short timespans. However, I opted not to do this in the paper, mainly because I didn't want to pick fights. There is a bit of a paradox here: If I claim that the conclusions in a paper claiming secular slip rate changes (or fluctuations) are actually due to aleatoric variability in earthquake recurrence, I will probably anger those authors and decrease the likelihood that they will consider these results in subsequent work. I'd rather write a more toothless paper that doesn't single out any given researchers, and is therefore a bit easier to swallow by all.

3. *Incorporating measurement uncertainty.* Measurement uncertainty is a very large factor that affects the results of any slip rate measurements, and I fully agree with comment [O6b] that it is in most cases underreported, both in offset measurements (as that comment references) and in geochronologic dating of any sort. I chose to leave it out of this paper because I really wanted to focus on the aleatoric variability, which is generally *underappreciated* as opposed to *underreported*.

Per a discussion of the physics and mechanics behind recurrence behavior: I have added a short discussion, but I don't want to really dig into the topic, for two reasons:

1. The intended audience for the paper is not only crustal deformation researchers, but others in the seismic hazard community as well–this includes engineers, geotechnical workers, analysts in the insurance industry, etc. In my experience as a member of this community, many others are only interested in these sorts of phenomena to the degree that they are consequential and actionable; their intellectual interests are often oriented towards their fields of expertise (structural engineering, ground motions, human and economic exposure, etc.). I want this paper to be a straightforward reference for how to evaluate slip rate data in light of aleatory variability that is not tied down in jargon or linked to specific geological or geophysical models or ideas that may not stand the test of time. Because of how variable and poorly-understood earthquake recurrence and fault interaction phenomena are, an in-depth discussion without resolution may well be off-putting to much of the audience that I would like to read this paper.

2. I don't think that we have a great understanding of the real mechanisms, yet. There are a variety of mechanisms under consideration (e.g., co- and post-seismic elastic and viscoelastic Coulomb stress changes, stress transients, dynamic triggering, pore fluid pressure fluctuations, fluctuations in the frictional failure threshold on a fault) in addition to actual secular changes in tectonic loading rates. The time-dependent mechanisms (particularly post-seismic processes) often show different behavior with regard to whether they are 'spun-up' and at a dynamic equilibrium, or not. And all of these mechanisms are necessarily linked to uncertainty as to how (and where) faults are loaded to begin with–whether the loading is in the elastic crust, in the viscoelastic/viscous mid or lower crust (in a continuum style), or on a discrete creeping dislocation down-dip of the brittle fault. There is a big range of scientific opinion on all of these questions. As a community we are begging for a big review paper to at least concatenate and organize these ideas and potentially test them or at least sort them into compatible vs. mutually exclusive sets for future testing. But we don't have that right now, so the topic is kind of a big mudhole. I will dip my toe in but I really want to avoid falling in for the purposes of this paper.

*Changes: New experiment with empirical slip distribution, and discussion of physical mechanisms behind aleatory recurrence variability.*

> [O5] There is a literature of ideas to draw upon, such as post-seismic fault reloading (Kenner and Simons, 2005), earthquake super cycles (Sieh et al., 2008; Weldon et al., 2004), isolated versus fault-network behavior (Berryman et al., 2012). Some of these ideas are discussed briefly but need more explanation.

I have added a short discussion (two paragraphs) on the topic, but as noted in my response to [O4] I don't think that a more full discussion is warranted. I want this

paper to be a simple, easy-to-digest paper and I think that a long and necessarily unsatisfying discussion (as we don't have answers yet) on the mechanisms behind recurrence variability will be an obstacle, and many readers will just put the paper down.

*Changes: discussion added.*

> [O6a] Likewise one could examine actual earthquake slip distributions (not landform offsets of historic events, which convolve landscape processes with tectonic slip) to develop an empirical basis for the slip function.

Such a distribution is given by Biasi and Weldon (2009), following work done by Hemphill-Haley and Weldon (1999). It is a bit different than the lognormal distribution used in that the probability of relatively low values (zero or near-zero) is higher than in a lognormal distribution. The sample COV of this distribution is 0.67, slightly lower than the lognormal slip distribution used in the paper, with a COV of 0.75.

As an experiment, I have re-done the simulation sampling from this distribution; the data are given as 1313 discrete points from earthquakes worldwide, normalized to the mean slip per event. I have sampled randomly from this finite set, with replacement, instead of interpolating the set into a continuous distribution and sampling from that. The results are in the new supplemental materials (Figures S1 and S2, Table S1, and some discussion).

The results are nearly indistinguishable.

*Changes: new experiment added.*

> [O6b] Some of the scatter in slip distributions is likely due to under-reported measurement uncertainty (Gold et al., 2013), and thus the cancellation of this error over multiple earthquakes should let cumulative slip converge more quickly than may be predicted from the author's model.

Either I don't understand this comment (which is quite possible) or it is a bit misapplied. The only reading of this comment under which one expects faster convergence than I have modeled is if the reviewer believes that measurement error is some component of the total variability represented. But that is not the case in the modeling; these distributions are taken to represent only aleatory variability and the study is performed with assumptions of zero measurement error.

*Changes: None.*

> [O7] Page 1, line 4. The open interval problem is well known and attempts to quantify it do exist on case-by-case basis.

Yes, and some of these cases are cited in the introduction. However, the open-interval problem simply deals with the uncertainty in a single recurrence interval (the present one), and not the variability that is present throughout all of the closed earthquake

intervals that have contributed to the measured offset; this larger issue is the topic of the manuscript.

*Changes: None.*

> [O8] Page 1, line 13. It seems odd to characterize uncertainty due to a random distribution as epistemic. Isn't this unreported aleatory uncertainty?

Aleatory and epistemic uncertainty are not mutually exclusive categories. Much epistemic uncertainty results from aleatoric variability, particularly when the underlying distributions that characterize the aleatoric variability are not known.

This is one of those instances: The framing of the situation is that one has made a single slip-rate 'measurement' (net slip / time) without knowledge of where the fault is in its earthquake cycle, what the past earthquake history is, and what the distributions of slip and recurrence are for that fault to begin with. Thus the condition is one of ignorance, i.e. epistemic uncertainty, and this section of the study shows how to approximately quantify this uncertainty under different assumptions of the slip and recurrence distributions.

*Changes: None.*

> [O9] Page 1, line 20: Why is marker in quotes?

I wanted to declare that it was a technical term and not a word that I arbitrarily applied to the situation. But this isn't necessary.

*Changes: Quotes removed.*

> [O10] Page 2, line 5. afterslip and creep also contribute.

Truth.

*Changes: Afterslip and creep added to sentence.*

> [O11] Page 2, line 10-11. Awkward sentence. Break into two.

Ok.

*Changes: Sentence broken.*

> [O12] Page 2, line 13 and other citations: persistent use of 'e.g' after citing only one or two articles is poor form and makes this reader think that the author has not adequately explored the literature.

The use of 'e.g.' denotes that the given citations are not authoritative or canonical in the sense that the cited works are where the concepts given are first introduced or best developed, as this isn't true. The cited works are generally just modern, high quality studies that exemplify the topic at hand.

I don't really care what readers may think of the depth of my scholarship.

For what it's worth, 'e.g.' should be before the references but was placed after by a LaTeX bug that I hadn't diagnosed.

*Changes: None.*

> [O13a] Page 3, line 11. This is not the correct definition of an exponential
> / poisson distribution. There is no prescribed number of events, only a
> prescribed time-independent probability.

The time-independent probability is the mean rate of events. The mean rate of events is the mean number of events that occur within some time interval.

Obviously with finite sample sets (of time, or of events) there will be some variation–otherwise I wouldn't have written the paper.

Nevertheless, the statement actually made is that the spacing between uniform random samples in some interval is characterized by an exponential distribution, which is true. It is not stated that this is the definition of the distribution.

*Changes: None.*

> [O13b] It is also worth noting that this is physically unrealistic at short
> time intervals because it violates elastic rebound.

Elastic rebound is a hypothesis, not a law, and is phenomenological instead of physical in nature. It is unfortunately a step removed from the modern understanding of the mechanics of earthquakes, which are based around stress, not strain. These map to each other nicely in the case of elastic and Newtonian viscoelastic rheologies, but not as nicely with rate-dependent rheologies, which are often considered the best characterization of the lower crust and upper mantle (e.g., Hetland and Hager, 2006). It's also hard to put strain in a framework with friction, for example. 'Physically realistic' modeling has to make a lot of assumptions and use heavy duty equipment (finite elements, for example) to incorporate strain.

There is also a separate issue with elastic rebound: It's not very easy to tell whether all the accumulated shear strain was released in an earthquake or not. What kind of measurements would tell us this?

I strongly suspect that we are fundamentally underestimating the frequency of very short recurrence intervals on faults. They're close to invisible to paleoseismology, which is our main source of data for recurrence interval statistics, because very closely-spaced events may not each produce differentiable colluvial wedges or other signs of surface deformation. This could plausibly result in a strong sample bias in the statistics. Nonetheless, we have clear observations of short recurrence intervals in the past few years. For example, some parts of the Monte Vettore fault slipped about 20 cm in the Amatrice earthquake and then ∼2 m in the Norcia earthquake a few months later (Gruppo di Lavoro INGV sul Terremoto di Amatrice-Visso. (2016, October 29). PRIMO RAPPORTO DI SINTESI SUL TERREMOTO DI VISSO ML 5.9 DEL 26 OTTO-BRE 2016 (ITALIA CENTRALE). Zenodo. http://doi.org/10.5281/zenodo.163818).

From a fault mechanics perspective, some researchers (for example Mark Zoback and his students, primarily Townend, as well as myself) believe that the shear stress on a fault required to initiate failure is much greater than the stress drop during the event, i.e. shear stress does not go to zero. Failure is decently described by

Mohr-Coulomb models, and at, say, 10 km depth, the confining pressure is almost 300 MPa. With a reasonable rock density (2700 kg/m$^3$), coefficient of static friction (say 0.5), and pore fluid pressure (say 0.3 times lithostatic pressure), the shear stress at failure is 94.5 MPa. Stress drops are generally on the order of 0.1-10 MPa (see Peter Shearer's work on Brune-type stress drop estimates, for example). So if less than 10% of shear stress is relieved in an earthquake, what are the implications for the elastic rebound hypothesis?

My take on this is that elastic rebound is a great way to describe the phenomenology of earthquakes to a non-geologist. Scientifically, it was an idea of absolute genius in 1910, but it isn't a thorough or mechanically sound framework for earthquake science a century later. Stress-based frameworks are much better suited to both conceptual and quantitative treatments.

*Changes: None.*

> [O14] Page 3, line 24. It would be useful to briefly discuss how shape and scale affect distributions generally. Shape governs the how tailed and is dimensionless; scale determines the spread of the distribution and is dimensioned (in years for this case).

Good idea. I added brief definitions of the scale, shape and location parameters to an earlier paragraph where the terms are first written.

*Changes: definitions added.*

> [O15] Page 4, line 6. Pareto distribution is another, simpler distribution needing only shape and scale to describe COV > 1

The Pareto distribution isn't used to describe earthquake recurrence, as far as I know. I believe that its only use in seismology is the tapered Pareto distribution for magnitude-frequency distributions by Yan Kagan (and perhaps others); this is not a similar-enough use to include here.

*Changes: None.*

> [O16] Page 4, line 15. Akciz et al (2012) revised Grant and Sieh (1994) and found much more periodic behavior.

The referenced sentence simply states "However, one can find examples of studies indicating the opposite conclusions", to reinforce the paragraph's opening statement that "No consensus exists among earthquake scientists as to the most appropriate recurrence interval distribution." That a study revised a previous study and found different results further reinforces this point, but I don't think there is additional benefit to citing it.

*Changes: None.*

> [O17] Page 4, line 28. The author should consider non-dimensionalizing the results of this study to facilitate more general use of its results. Instead of mean slip of 1m, one would refer to non-dimensional slip of 1 and multiply by average slip per event to scale the results. This is

> effectively what the author describes already, though without formal non-dimensionalization.

I considered formal nondimensionalization but decided against it.

In the end, I decided to present the results dimensionally because most geologists think dimensionally (myself included).

Dimensional thinking allows for different heuristics to be used when analyzing methods and results, than non-dimensional thinking. Basically, non-dimensionalizing parameters forces parameters to only be defined in terms of their relationship with each other, and to only exist within the context of this specific problem.

If I say a fault has a mean slip of 1, and a mean recurrence interval of 1, and a mean slip rate of 1, it's hard to picture such a fault. Are those reasonable values for these parameters, individually? Can't say. Furthermore, the slip rate isn't even 1, it's 1/1, because it's a rate and the non-dimensional units are still not arithmetically compatible units–multiplication and division are possible (kind of, but not reducible) but addition and subtraction are not at all defined. (Please forgive me for not knowing enough measure theory to rigorously state this...).

Non-dimensionalization in many geological problems reduces the clarity of the analysis or solution because it strips it of context. At the same time it can facilitate the manipulation of the equations within the study, or during programming, etc. Physicists like it because it makes their work easier, I would say, and they're all pretty used to it.

Compare this to the dimensionalized problem in the paper. Can you imagine a fault with a mean per-event slip of 1 m, a recurrence interval of 1000 years, and a slip rate of 1 mm/yr? I can, and I can place it–it's a small but pretty active intraplate fault, or a splay in a plate boundary. It's the kind of fault one might actually go and trench.

However, all the numbers are 1 (or 1000) so that it is easy to scale to other faults by multiplying by non-dimensional scaling factors (i.e., for a Mongolian fault with a slip rate of 3 mm/yr and a per-event slip of 10 m, you scale the slip by 10 and the slip rate by 3). It's not accidental that I have chosen these numbers, and I have explicitly described how to do the scaling.

*Changes: None.*

> [O18] Page 5, line 15. The statement 'appears to be related to COV' is disappointing. Given that this paper is entirely simulation, the author should be able to make a quantitative comparison of slip-rate variance to COV.

Sorry to disappoint.

A comparison between COV and the slip-rate variance at any time is actually done in the study (it is in fact the core of the study)–I vary the COV of a single distribution (the lognormal distribution) while changing nothing else, then I keep the lognormal

distribution with a COV of 1 and compare that to a very different distribution (the exponential distribution) with a COV of 1. The results, which are described unambiguously in the study, show that the slip rate variance changes with COV but isn't much affected by the shape of the distribution (i.e. lognormal vs. exponential).

There is no reason to pursue this farther here. The work that I have done here has covered the (small) space of geologically reasonable distributions; there aren't huge gaps left unexplored.

There are two main families of distributions that are in consideration for earthquake recurrence: Exponential distributions (possibly with modifications such as a stretched or hyperexponential distribution) and lognormal-like distributions (lognormal, Wiebull, BPT, etc., which are not distinguishable in real paleoseismic datasets). I have compared both types of distributions for a single COV and the differences are very minor. There are no alternate distribution families in any consideration within seismology that are *more* different than these two families.

This work is not meant to offer a mathematical proof, which is why I gave the soft 'appears to be related to...' statement instead of a more firm 'is demonstrated to be caused by...' or 'is proven to be a function of...'. Demonstrating some correlation or relation is enough to further the general point, which is that more earthquake recurrence variability will lead to more short-term slip rate variability (which seems self-evident in any case).

Going farther would either mean doing many more numerical experiments (which do not offer real proof and would just clog the paper) or invoking more complicated mathematical tools such as stochastic calculus, which I don't really know and am unwilling to teach myself for this paper.

*Changes: None.*

> [O19] Page 5, line 29. This problem has been studied (Weldon et al., 2004; Sieh et al., 2008). The Sumatran subduction zone work is particularly relevant and completely overlooked here.

The work by Weldon et al. (2004) is referenced on Page 6, Line 3. Neither of these papers deal with the topic of autocorrelated recurrence intervals in any quantitative or otherwise explicit manner, and I don't read anything that I can interpret as a qualitative discussion either. One can read both papers and not get a sense of whether a short recurrence interval implies the next recurrence interval will be short or long, much less any quantification. Both papers deal with the concept of 'earthquake supercycles' or groups of earthquakes that are relatively tightly-spaced and separated from other groups by long recurrence intervals. This may share a conceptual link, but from a technical perspective this likely has more to do with periodicity than autocorrelation, and these are mathematically separate such that a periodic sequence may have positive or negative autocorrelation. Goh and Barabasi (2008, Europhysics Letters) is a useful discussion on the topic of autocorrelation vs. periodicity in regards to quantifying clustering behavior (though one may safely ignore the references to seismicity in that paper).

[Note: I've analyzed Weldon's data (from K. Scharer's refined OxCal earthquake dates) and found that earthquake recurrence interval duration at the Wrightwood and Pallett Creek sites has a negative autocorrelation, i.e. a short recurrence interval is likely to be followed by a long recurrence interval, and vice versa. This is quite unlike the autocorrelation in the Puget Lowlands of Washington State, which is a network of generally similar faults; here autocorrelation is positive, so a long recurrence interval will more likely be followed by another long recurrence interval. I don't know what this means, or whether it's all noise, but it's intriguing to me and I put this bit in the paper in hopes of catching the attention of others who may be looking for a project.]

*Changes: None.*

> [O20] Page 6, line 1. Zero friction at rupture arrest is very unrealistic, and not a prerequisite for characteristic behavior.

It is not a necessary condition but it is a sufficient one, when coupled with fairly regular reloading and failure conditions. I personally think it's unlikely as well (see Styron and Hetland 2015 for example), but zero (or very near zero) friction is part and parcel with complete stress drop in major earthquakes, which is supported by many studies (e.g., Hardebeck, 2012; Hasegawa et al., 2011) though I am pretty suspicious of the results. Furthermore, there are a host of laboratory experiments which suggest that friction during slip decreases to very low values (e.g., Di Toro et al., 2004).

*Changes: None.*

**Review by Anonymous Reviewer #3**

> [B1] Uncertainty in estimates of earthquake recurrence and fault slip rate are important parameters, pursued by conscientious investigators of seismic hazard. The author perceives a lack of statistical support, and offers in this paper "insights" in variously tones of "friend of the practitioner" and "trust me, I'm the numerate one here". Neither is convincing.

Someone's grumpy!

Since this review seems to aim to shame rather than help, the number of changes are few.

> [B2a] A couple of omissions in this paper are particularly striking.

> [B2b] First, how do we have a paper addressed to "variability" in fault slip rate, addressing particularly the problem of small samples, without mentioning the methods of estimation for censored samples? There is an extensive statistical literature to estimate parameters and uncertainties and numerous recent papers applying it in paleoseismic contexts. This literature provides real quantitative methods to deal with the open

intervals, long or short, that affect the geologist's estimate of fault slip rate and recurrence estimation. These are real equations, with real uncertainties. One would look in vain in this paper for anything of similar substance.

The paper doesn't go into depth on the topics of open intervals and censored data because this paper does not use any samples of earthquake recurrence intervals or per-event displacements in the rate calculations. Though they are present behind the scenes, all of the numerical results in the paper are slip rates derived as the total offset over the total time (equation 1)–no information on how many earthquakes are present in that time window, or what the duration of open intervals may be, is used in the calculations. This is intended to simulate a neotectonic-type slip rate estimation (as written in page 2, lines 24-25 of the original manuscript), where an offset geological marker layer is dated and the offset is measured. I have reinforced this by restating in the methods that neither the number of events or the length of the closed or open intervals are known.

The decision to avoid discussing the topic of including censored data in parameter estimation is also warranted because the most robust method to incorporate open intervals are in maximum-likelihood estimation, which is a complex topic to bring up, as are survival analysis and related methods. Since these topics are complimentary, not central, to this paper I don't see the benefit of adding a dense paragraph (much less any equations) for methods that aren't relevant.

I don't disagree that more substance is offered by methods developed through the decades by many researchers than I can offer in one short paper. Sorry!

*Changes: brief reinforcement that this study uses neotectonic (age/offset) slip rate measurements, not those from multiple events as in a paleoseismic context.*

> [B2c] Second, pages of this paper could be replaced (and improved) by a presentation and discussion of the properties of the standard error. E.g., given an estimate a sample- based estimate of the mean, how far might the population (or true) mean be from the estimate? S.E. is estimated by the sample standard deviation divided by the square root of the number of samples. So, of course, estimates from small samples from a fuzzy log normal converge more slowly than from a well-defined (quasi-periodic) lognormal. Instead of a small equation (SE=s/sqrt(n)), our paper back-calculates the result using 2 million years of samples, and presents the results like a new discovery . And again, with little by way of meaningful uncertainties (e.g., p1, lines 11- 13, 14- 16).

The standard error does share the same basic decrease in dispersion with increasing number of events, but it is symmetric even though the distributions used here are not, and it does not account well for the systematic bias in short time windows that are observed in these results. The misfit between the symmetric and asymmetric approximations becomes particularly bad when estimating the 'epistemic' uncertainty as in Figure 6, which is the uncertainty that one should apply to a measurement to account for earthquake-cycle variability. I have calculated the standard error and

added to this graph show in Figure 2 below (not added to manuscript). The results do not support the use of the standard error here. The upper error envelope under-approximates the uncertainty at short time windows (due to the measurement bias), but then becomes acceptable. However, the lower error envelope stays at or below the 5th percentile, though it should be the 1-standard deviation approximation.

[Figure]

Figure 2: Standard error (S.E.) plotted on top of Figure 6 from the manuscript.

The standard error is also not a great fit for neotectonic slip rate estimations: They are typically derived by dividing net offset over net time $n$ is not known, nor are the durations of the open intervals. It would be different if this paper was at all concerned with the calculation of recurrence intervals, per-event displacements, or other parameters of individual earthquakes where a number of samples is used, but that is a different topic.

Finally, meaningful uncertainties are given in Table 1 for all distributions at a wide range of times. It's not appropriate to reduce all of these to a single equation.

*Changes: Short discussion of standard error added to 'slip rate calculations' section.*

> [B3] 1 L 9,10: We read that the most important parameter is the co-
> efficient of variation. First, this equation is the arithmetic coefficient
> of variation, and not the CV for a lognormal distribution. The CV of a
> lognormal does not depend on the mean.

The reviewer is really confused about the lognormal distribution (see [B6] and [B7] where the reviewer continues to make mistakes and level false accusations at me).

I was quite worried that I made a mistake after reading this, so I re-checked my math, my code and re-read a range of references (e.g., *Statistical Distributions* by Evans, Hastings and Peacock, 1993; *Confidence Intervals for the Coefficent of Variation for the Normal and Lognormal Distributions*, Koopmans, Owen, and Rosenblatt, *Biometrika*, v. 51, no. 1/2, 1964, as well as a lot of internet pages). Nowhere do I find alternative definitions of the $CV$ for a lognormal distribution, except in the Wikipedia page for Coefficient of Variation where two conflicting definitions for the Geometric Coefficient of Variation are given. Both of these involve estimating the $CV$ from samples rather than as an algebraic transformation of the equations for a lognormal distribution. The first definition is mathematically the same as the standard definition; the difference lies in whether the estimation procedure is done on the raw data or log-transformed data. The second is simply the square of the first.

Just to reiterate, the arithmetic coefficient of variation for the lognormal distribution is the standard definition. It is also equivalent to the geometric coefficient of variation for a theoretical longormal distribution (although not necessarily for sample-based estimates).

The reviewer's confusion may stem from the fact that there are two equations for the coefficient of variation that get regularly passed around. However, as I demonstrate in the response to [B7c], they are mathematically equivalent. The simple definition, which I use in the paper, is $CV = \frac{\sigma}{\mu}$, where $\sigma$ and $\mu$ are the standard deviation and mean of the distribution. The other definition, $CV = \sqrt{e^{\phi^2} - 1}$, is given in terms of the standard deviation $\phi$ of the natural log of the lognormal distribution.

OK, this is pretty funny: All of the confusion that the reviewer has about the log-normal distribution appears to be because of a misreading of the Wikipedia pages for the lognormal distribution and for the coefficient of variation. I mentioned the geometric CV (non-)issue above. The quip that the 'CV of a lognormal does not depend on the mean' is… kind of true? As demonstrated in the response to [B7c], the standard deviation of a lognormal is a *function of the mean* in that the mean is a coefficient in the standard devation equation if it's expressed in the typical shape and scale parameters. Therefore $CV = \sigma/\mu = \mu A/\mu$ where $A$ is another mathematical expression (see [B7c]). So, yes, technically it is true but only because $\mu$ is on the top and the bottom of the fraction and therefore cancels out; furthermore, you can't derive the expression $A$ without knowing both $\mu$ and $\sigma$. Regardless, the quip gives all outward appearance of being a slight modification of the sentence "Contrary to the arithmetic standard deviation, the arithmetic coefficient of variation is independent of the arithmetic mean" which is below the equation for the $CV$ of a lognormal on

the Wikipedia page. It is not a statement that makes any practical sense unless one is beginning with log-transformed data or distributions.

So to conclude, my math, code and analysis are correct in terms of the use of the $CV$.

*Changes: None.*

> [B4] We could stop here, but a central flaw in the paper is exposed – nothing in this paper addresses how to obtain this most important parameter. If attempted, the essential emptiness of a 2,000,000 year sample would emerge. No real data set in paleoseismology resolves the mean and standard deviation to better than maybe 50%. Typical sites do well to resolve it to a factor of 2. P.3, line 22- 23 reflect this reality.

"the essential emptiness". . . I'm surprised, and perhaps a bit concerned, to find that I have tapped into the existential despair of Anonymous Reviewer #2.

This study investigates how slip rates measured as *offset / time* of an offset marker are influenced by earthquake cycle variation. It is *not* a paper discussing methods or uncertainties invovlved in the calculation of the mean and standard deviation of earthquake recurrence intervals or displacements from paleoseismology. Indeed, if "There is an extensive statistical literature. . ." (comment [B2b]) on this topic, why should I rehash it?

I fully agree that a typical paleoseismological study cannot resolve the mean and standard deviation with a high degree of accuracy and I never stated otherwise. Partly for this reason, by using CVs of 0.5, 1, and 2, I cover a pretty wide range of values that can give some idea of the expected behavior of the almost any fault system if the basic recurrence distributions can be estimated or assumed.

A discussion of 'guesstimating' the $CV$ is actually already present in the manuscript, starting on p.8, l. 31 in the original submission. However, I do not give an in-depth treatment of estimating the $CV$ on a fault–this topic would be suitable in a paper on interpreting field methods or a general overview of statistical paleoseismology, perhaps, but not in a numerical simulation paper that takes values of the $CV$ *a priori*.

*Changes: None.*

> [B5a] It is not obvious that the author has material experience words "aleatoric" and "epistemic". Line 1, "aleatoric uncertainty" is a contradiction in terms.

It's really not a contradiciton in terms. 'Aleatory' does not mean 'the absence of uncertainty', which it would have to in order to contradict the term 'uncertainty'.

In any case, the term 'aleatoric uncertainty' is commonly understood and used by statisticians in addition to us proles. See for example O'Hagan, T. (2004). Dicing with the unknown. *Significance*, 1(3), 132-133.

*Changes: None.*

> [B5b] Bird, Zechar and Frankel all know better than to use the method
> the author alleges in lines 21 and 22 to arrive at epistemic uncertainty in
> slip rate. They would more likely consider the allegation a misreading
> of their work.

This is a very perplexing comment. The line in question states, "The uncertainty in the resulting slip estimate is typically treated as epistemic, and quantified through the propagation of the measurement uncertainties on the offset and time quantities (e.g., Bird, 2007; Zechar and Frankel, 2009)." Both of the papers referenced are focused on estimating fault slip rates (or *long-term offset rates* in Bird's terminology) that incorporate the measurement uncertainty in both age and offset measurements. Both of their methods are based in a convolution of the probability distributions for age and offset of the marker features (i.e., the data with their uncertainties). Perhaps the reviewer takes 'the propagation of the measurement uncertainties...' to mean that it is done through adding them in quadrature to the final rate, but I clearly do not specify the methods used here and the reviewer may want to reign in the assumptions.

*Changes: None.*

> [B5c] More broadly, the lack of care in writing makes one wonder how
> to understand this paper.

Sorry!

*No changes.*

> [B5d] p2, L5: A perturbation in slip rate would mean it was slipping at
> rate X, then changes to Y.

Yes. This is both the intended and the obvious interpretation of the sentence. The slip rate at the fault trace varies because of variability in the frequency and magnitude of accumulating offsets.

*No changes.*

> [B5e] p.2, L14-17 have careful paleoseismologists doing reasonable
> things in one sentence, then imply they would make plainly rookie
> mistakes in the next.

The paragraph in question states that the open interval since the last earthquake is often taken into consideration, however the variability in closed intervals is not nearly as often taken into consideration–this is why I have written the paper. Should I not state that paleoseismologists and neotectonicists who consider the effects of the terminal open interval are careful? I don't feel like being less generous. Should I not point out that variability within the closed intervals is important and rarely considered? No, I think that I should. The first reference in this paragraph makes both the 'reasonable' decision and then the 'rookie mistake'. I am not simply making this stuff up.

*Changes: None.*

[B5f] From here these read like inexperienced generalizations.

Sorry!

*No changes. Changes: None.*

[B6a] p.4, L15-23: The descriptions of the lognormal variables here give one pause. First, log-normal parameters do not have units.

This comment is completely off the mark. The shape and scale parameters of a lognormal distribution are indeed unitless, as they are the mean and standard deviation of $\ln L$, where $L$ is a lognormal distribution (this transforms the lognormal distribution into a normal distribution).

However, the mean and standard deviation of the actual lognormal distribution $L$ are in the units of the distribution itself. Nowhere in the manuscript did I state that $\mu$ and $\sigma$, which are clearly defined as the mean and standard deviation of any distribution in question, are the shape and scale parameters of $L$. They are simply the mean and standard deviation of the distribution.

I have added a footnote clarifying this and giving the derivation of the shape and scale parameters from the mean and standard deviation.

*Changes: Footnote with explanation and equations added.*

[B7b] Second, the mean recurrence interval is not the location parameter of a log normal. This is just wrong.

Nowhere in the manuscript did I state this. In fact, the word 'location' is not present anywhere in the submitted manuscript (it has been added to an expanded discussion of probability distributions and their parameters in the revised manuscript as described in the response to Comment [O14], though I still do not make the mistake that I have been accused of).

Furthermore, the lognormal distribution doesn't have a location parameter–it is a 2-parameter distribution specified by shape and scale parameters, as stated in P. 3, L. 24 of the submitted manuscript.

*Changes: None.*

[B7c] Third, if one uses the CV equation for the lognormal distribution (e.g., https://en.wikipedia.org/wiki/Log-normal_distribution#Arithmetic_coefficient_of_variation), the CV will not match the COV alleged here. Given that the study depends on these distributions, we can't really use subsequent conclusions.

This is false. The equations are mathematically equivalent.

The mean $\mu$ of a lognormal distribution $L$ is often given in terms of the mean $m$ and standard deviation $\phi$ of the log-transformed distribution $\ln L$, which is a normal distribution. The equation is:

$$\mu = e^{m + \frac{1}{2}\phi^2}$$

and similarly, the standard deviation $\sigma$ of $L$ may be given in terms of $m$ and $\phi$:

$$\sigma = e^{m + \frac{1}{2}\phi^2}\sqrt{e^{\phi^2} - 1},$$

which is clearly equivalent to

$$\sigma = \mu\sqrt{e^{\phi^2} - 1}.$$

Therefore,

$$CV = \frac{\sigma}{\mu} = \frac{\mu\sqrt{e^{\phi^2} - 1}}{\mu} = \sqrt{e^{\phi^2} - 1}$$

which is the equation for the $CV$ of a lognormal distribution given in any source including the Wikipedia page. Please note that in most sources, the symbols '$\mu$' and $\sigma$' are what I have here defined as '$m$' and '$\phi$' because I use the symbols $\mu$ and $\sigma$ to represent the mean and standard deviation of $L$, consistent with the manuscript where the same symbols are used equivalently across distributions.

The methods and results of this study are consistent and this alleged error did not occur.

*Changes: None.*

> [B7] p.6, L20. If the number of samples is really n = N – t + 1, the samples are correlated by virtue of the overlap in the windows. No accounting has been made of the correlation structure.

There is certainly a lot of autocorrelation of the samples if one were to look at them sequentially. Nonetheless, even if sequential samples are correlated, given enough samples, the final distribution of the samples won't depend on the autocorrelation in the samples (this principle underlies the family of powerful and commonly-used numerical methods based on Markov Chain Monte Carlo sampling, for instance). This is why I use 2,000,000 years of simulation: it ensures that the sampling is ample enough to not worry that the serially-correlated samples have not sufficiently sampled the distribution. I would have to use a much longer simulation (tens to hundreds of millions of years) otherwise.

*Changes: None.*

> [B8a] p.6, L26: Starts a narrative of the consequences of the standard error, as though the standard error was never invented.

I have added a brief discussion of the standard error, which is an OK approximation of the results beyond 5-10 mean earthquake cycles (once the systematic bias in very short time windows has gone away), but the asymmetry of the results are not well approximated by the standard error. One could hack on it a bit by log-transforming the data, applying the standard error of the log-transformed distributions, and then anti-log transforming it, but that seems a bit much.

*Changes: None.*

> [B8b] The fuzzy, back-of-the-envelope estimates start to get thick here. Real uncertainty estimates would serve better.

The uncertainty estimates at a range of percentiles are available in Figure 5, and the uncertainties for a slip rate measurement are in Figure 6 and Table 1.

*Changes: None.*

> [B9a] p.7, L1-3: Two observations: First, as written, the practicing geologist is being asked to believe that 60 earthquake cycles have passed with zero displacement. I can guess what was intended, but should not have to.

Actually, the naive reading of the sentence, that 60 *mean* earthquake cycles passed without an earthquake on a fault in the simulation, is exactly what is intended.

*No changes.*

> [B9b] Second, what probability is associated with this 60-cycle thing? I ask because practicing hazard geologists have to make estimates, and give weights to extreme events. What is the probability of 60 cycles, a CV of 2.0, . . . ? Hard to imagine that the author has thought much about what these results would mean or how to use them if they were true.

I happen to be a practicing hazard geologist who is responsible for the implementation of fault sources for probabilistic seismic hazard analysis, so I have actually considered this. It's very important when deciding whether to include faults that have very clear bedrock geomorphic signatures (say, from rapid deformation in the Miocene) and are favorably oriented for slip in the current stress regime, but lack evidence of Quaternary deformation. Are they to be included in the source models, or not? I tend to add them, with low slip rates and high uncertainties, for just this reason. The analysis in this paper shows that very long recurrence intervals are possible on active faults.

There are two probabilities that are asked here: 1. *The probability of a fault with a CV of 2.0 going 60 mean earthquake cycles without accruing much displacement.*

This is answered very easily using the equation for the lognormal probability distribution with a *CV* of 2. The cumulative distribution function for this recurrence distribution at 60,000 years (60 mean earthquake cycles) is 0.9994..., which means that there is a probability of about $p = 0.0006$ that any given recurrence interval will exceed 60,000 years, or 60 mean earthquake cycles. But the probability $P$

of a recurrence interval this long being present in a sequence of $n$ earthquakes is $P = 1 - (1 - p)^n$, and for 2000 earthquakes (which is the approximate number in these simulations) the probability of a single recurrence interval exceeding 60 mean earthquake cycles is 0.1 or 10%. It becomes more likely than not that such a recurrence interval will be present in a sequence at about 12,500 earthquakes. Are you going to see this in a trench? Almost certainly not even if you were to have the appropriate sedimentation to preserve the signal. But that does not mean that it's an impossible or even improbable event in a long time series.

*Changes*: I have added the following sentence to the paragraph under discussion: "It is highly unlikely that any given recurrence interval will be this long, but given thousands of earthquakes over millions of years, the chance of such an event occurring at least once is far more likely." I don't find it necessary to go through the probability calculations in the manuscript as I did here.

2. *The probability of a fault having a CV of 2.0.*

The best way to calculate this probability is basically to look at the frequency of such a CV occurring in some sample of faults. I don't have a big collection of faults with paleoseismic data to work with, and I'm not going to put one together to satisfy a reviewer, but I do calculate that the schematic, representative model for southeastern Australian intraplate faults by Clark et al., 2017 (cited in the paper) has a CV of 2. Unfortunately, Clark et al. cannot precisely date all of the earthquakes they infer so no real CV can be calculated from their data, but this is *their* best guess as to how the faults behave, based on their evidence for clusters of events with recurrence intervals of estimated mean 8,000 years separated by quiescent intervals of 0.5-2 million years.

Similarly, the USGS Quaternary Faults and Folds dataset has a number of faults that are thought to have last ruptured in the 'Middle and Late Quaternary' (<1.6 Ma) but not in the 'Late Quaternary' (<130,000 years). Disregarding potential misclassifications of these faults, it seems reasonable to assume that a few of them may be experiencing very long recurrence intervals and may have a CV approaching 2. Many of these faults are along strike of fault segments with Holocene ruptures and there is no obvious geological difference between the segments with recent rupture and those without–the Steens fault zone in the lovely Steens rift of southeast Oregon is an example.

*Changes: None.*